# Effects of Biostimulants on Annurca Fruit Quality and Potential Nutraceutical Compounds at Harvest and during Storage

**DOI:** 10.3390/plants9060775

**Published:** 2020-06-20

**Authors:** Giulia Graziani, Alberto Ritieni, Aurora Cirillo, Danilo Cice, Claudio Di Vaio

**Affiliations:** 1Department of Pharmacy, University of Naples Federico II, Via Domenico Montesano 49, 80131 Naples, Italy; alberto.ritieni@unina.it; 2Unesco Chair for Health Education and Sustainable Development, 80131 Naples, Italy; 3Department of Agricultural Sciences, University of Naples Federico II, Via Università 100, 80055 Portici, Italy; aurora.cirillo@unina.it (A.C.); danilo93mhr@hotmail.it (D.C.); claudio.divaio@unina.it (C.D.V.)

**Keywords:** annurca, biostimulants, polyphenolic compounds, hydroxycinnamic acids, flavonols, procyanidins, antioxidant capacity

## Abstract

The cultivar Annurca is an apple that is cultivated in southern Italy that undergoes a typical redding treatment and it is appreciated for organoleptic characteristics, high pulp firmness, and nutritional profile. In this study, the effects of three different biostimulants (Micro-algae (MA), Protein hydrolysate (PEP), and Macro-algae mixed with zinc and potassium (LG)), with foliar application, on the quality parameters of Annurca apple fruits at the harvest, after redding, and at +60 and +120 days of cold storage were analyzed: total soluble solids (TSS) content, total acidity (TA), pH, firmness flesh, and red coloration of epicarp. Additionally, the polyphenolic quali-quantitative profile of pulp and peel was analyzed by UHPLC-Q-Orbitrap HRMS and Folin-Ciocalteu and the antioxidant capacity with the methods 1,1-diphenyl-2-picrylhydrazyl (DPPH) and ferric-reducing antioxidant capacity (FRAP). The results obtained suggest that biostimulants are involved in the regulation of the secondary metabolism of the treated plants, acting positively on the quality of the Annurca fruits and their nutritional value. Fruits treated with PEP have shown, during cold storage, a significantly higher content of total polyphenols in flesh and a higher concentration of phloretin xylo-glucoside and phloridzin (350.53 and 43.58 mg/kg dw respectively). MA treatment caused, at the same time, an enhancement of flavonols between 0.6–28% and showed the highest total polyphenol content in the peel after 60 and 120 days of cold storage, with 2696.048 and 2570.457 mg/kg dw, respectively. The long-term cold storage (120 days) satisfactorily maintained phenolic content of fruits deriving from MA and PEP application, in accordance with data that were obtained for peel, showed an increase of 7.8 and 5.8%, respectively, when compared to the fruits cold stored for 60 days. This study represents the first detailed research on the use of different types of biostimulants on the quality of the Annurca apple from harvest to storage.

## 1. Introduction

Malus pumila Miller cv. Annurca, is one of the most important cultivars in Souther Italy, especially in Campania region, which accounts for 5% of Italian apple production. This cultivar is the only one to have received from the European Council the “Protected Geographical Indication”, an official designation related to the preservation of local and characteristics agriculture commodities [1]. This fruit is famous for crispness, special taste, and long shelf life [2,3].

Numerous studies in the literature report positive effects on human health of the polyphenolic compounds that are present in the Annurca apple [4,5,6,7].

Annurca polyphenolic extracts avoid the damage of gastric epithelia in vitro and rat gastric mucosa in vivo [8]. Annurca has also displayed anti-radical, hypolipidemic, and hypoglycemic activity [9,10,11], and it has been shown to reduce cell viability in colon cancer and HL-60 cell lines by activating apoptosis [8,12].

Together with the containing of plant vigor [13], one of the major problems of the Annurca cultivation is the insufficient redness epicarp at harvest, so much so that it is necessary to complete the redness of the fruits in “melaio”, where the apples are exposed to the sun, on a layer of straw, and they are manually rotated to complete the coloring [14]. The exposure of the fruits to the sun serves to favor the oxidation of the anthocyanins present in the peel, which becomes bright red in color.

The redding process lasts 20–30 days after the harvest, depending on the weather conditions [15] and it involves high production costs, reducing the competitiveness of the product on the national market. The Annurca apple shows a high shelf life after redness, maintaining a high firmness and its peculiar organoleptic characteristics [16]. The red color of the apple derives from chemical components, called anthocyanins, which belong to the flavonoids class. The accumulation of anthocyanins is influenced by environmental factors, such as light, temperature, and nutrition, as well as by genetic factors. With maturation the chlorophyll of the chloroplasts in the fruit epicarp, they undergo a degradation [17]. Maturation involves the synthesis of new pigments: the flavonoids, located in the vacuoles, of which the most abundant are anthocyanins. Anthocyanins are synthesized through the phenylpropanoid pathway whose precursor is phenylalanine. The first enzyme to act on this precursor is phenylalanine-ammonium-lyase (PAL).

The regulatory mechanisms of anthocyanin biosynthesis are important, because skin redness is an important factor in market acceptance for many apple cultivars. Bio-stimulants are composed of biological substances and microorganisms containing bioactive compounds as mineral nutrients, humic substances, vitamins, free amino acids, chitin, polysaccharide, and oligosaccharides [18,19].

According to a recent EU Regulation, plant biostimulants are defined mainly based on their activity, therefore biostimulants include various organic and inorganic substances (humic acids and protein hydrolysates), but also prokaryotes (e.g., plant growth promoting bacteria) and eukaryotes, such as mycorrhiza, N-fixing bacteria, and macroalgae [20,21,22].

Several literature studies highlight the important role of biostimulants in improving the efficiency of the plant’s metabolism, increasing plant tolerance to and recovery from abiotic stresses, facilitating nutrient assimilation, translocation and use, enhancing quality attributes of produce, including sugar content, color, rendering water use more efficient and enhancing soil fertility, particularly by fostering the development of complementary soil micro-organisms.

Studies reported in the literature showed the important role of biostimulants also in improving the coloring efficiency of fruit peel [23], showing the positive effect of the Sunred biostimulant (protein hydrolysate) and abscisic acid in the accumulation of anthocyanins in the “Red Globe” grapes. The objective of this study was to investigate the effects of different commercially available, foliar applied biostimulants on yield, fruit color, nutritional quality, and storability of Annurca apples.

## 2. Results

### 2.1. Effect of Biostimulants on Fruit Colour

ANOVA on L*, a*, and b* showed a high statistical significance between the biostimulants (B) and fruit exposure (E), less or not significant (ns) was for the interaction B × E (Table 1).

At harvest, after application with PEP and MA (Table 1), there is a greater redness skin and, therefore, a higher value of the color coordinate of the red “a” in all of the exposures considered: shade, intermediate, but above all in the sun exposure. Fruits treated with PEP show “a” by −1.44 in the shaded part, which increases until reaching the maximum of “a” equal to 27.15 in the face exposed to the sun. Additionally, with the application of MA, the fruits show a greater red color, in the shade with “a” equal to 0.59, which increases up to 27.90 for the part exposed to the sun.

The greater efficiency of the treatments with the PEP and MA biostimulants is also evident in Figure 1 in which an index colour (IC) at the higher harvest is observed as compared to the other two theses, in all 4 of the exposures analyzed, but especially in the sun part with associated values of 39.78 and 50.71. Additionally, in the IC values, after “melaio” (Figure 2), it shows greater fruits coloring after application with the 3 biostimulants when compared to the Control; in fact, in all of the exposures of the fruit, there is a very homogeneous red skin coloration.

### 2.2. Effects of Biostimulants on Total Soluble Solids (TSS) Content, Total Acidity (TA), pH and Flesh Firmness

An increase in TSS from harvest to fridgeconservation of about 20% was observed; this increase is strengthened while using LG and PEP biostimulants to +120 days of fridge conservation with brix° of 14.10 and 13.27. At harvest (Table 2) there is a higher acidity for all four of these, with average values equal to 9.5 g/L which decrease about 40% from after “melaio” to the fridge-conservation.

At harvest, for all 4 theses, the firmness showed values of around 5 kg/cm^2^, to then decrease after “melaio” around 3 kg/cm^2^ until reaching on average values equal to 2.5 kg/cm^2^ during the fridge-conservation at +120 days. Another parameter analyzed in the present study is the variation in pH which does not show particular differences in all 4 theses; the lower values are observed at harvest with an average pH of 3.36, which gradually increases up to the maximum average values of 3.80 at +120 days of fridge-conservation.

### 2.3. Effect of Biostimulants on Quali-Quantitative Polyphenolic Profile

The influence of three different commercially available biostimulants, including microalgae (MA), protein hydrolysate (PEP), and macroalgae mixed with zinc and potassium (LG), on the quali-quantitative profile of polyphenolic compounds, in the flesh and skin of annurca samples, is included in Table 3. Table 4 lists the identified compounds, their retention time, and exact mass spectra data. By integration of MS and MS/MS spectra, a total of 18 different phenolic compounds were identified.

Literature data were also used for the comprehensive evaluation of phenolic compounds. Single phenolic compounds were quantified while using calibration curves built with appropriate reference compounds. The investigated compounds were grouped in five classes, such as hydroxycinnamic acids (chlorogenic acid, coumaroyl quinic acid, and caffeic acid), flavanols (catechin and epicatechin), flavonols (rutin, hyperoside, kaempferol-3-O-glucoside, apigenin-7-O-glucoside, and isorhamnetin derivatives), procyanidins (procyanidin B1, B2, trimer, and tetramer), and dihydrochalcones (phloretin, phloretin-2-O-xyloglucoside, and phloridzin).

In this study, we have analyzed the UHPLC-HRMS profile of the polyphenols in both flesh and peel of ‘Annurca’ apple, in particular, taking into account the peculiar post-harvest storage of this fruit, the effects of the foliar application of the biostimulants were assessed at harvest time (unripe fruit), after reddening in “melaio”, and at different times of cold storage (60 and 120 days at 2 °C). The amount of total phenolics calculated from the data obtained by UHPLC-HRMS analysis was slightly lower and not well correlated from that estimated by the Folin–Ciocalteau method (Table 5).

For these reasons, we have considered Folin results and UHPLC-HRMS data to evaluate, respectively, quantitative and qualitative biostimulant effects. With regard to the peel, the concentration of the extractable polyphenols remained unchanged during the reddening–ripening process for control and for fruits deriving from the foliar application of the micro-algae biostimulant, whereas PEP and LG biostimulant treatments caused an increase equal to 2.4% and 7.4%, respectively. Therefore, the application of PEP and LG bistimulants determined a positive effect on polyphenolic total content of fruits ripened in melaio which was significantly higher than the control and microalgae treated fruits (+18% and +9% respectively).

This result was mainly due to the large abundance of hyperoside, in PEP treated fruits, the concentration of which was higher than that found in the other samples (517.19 vs. 468.18, 333.74 and 464.43 mg/kg dw, respectively, for control, LG, and MA treated apples). Moreover, the qualitative results showed that LG application caused a higher peel concentration of chlorogenic acid when compared to the other samples analyzed (545.38 vs. 476.68, 488.00 and 505.70 mg/kg dw, respectively for control, PEP, and MA treated fruits). Bearing in mind the lack of data on the effects of biostimulants on the nutritional quality of apples (flesh and peel) during cold storage, in this study we assessed how the qualitative and quantitative profile of polyphenols was influenced by the different treatments, during the storage period of fruits.

In particular, the control fruits increased their polyphenolic content by 28%, while the fruits derived from biostimulant treatments showed an increase of 17.5% (PEP), 25% (LG), and 28.7% (MA), and not significative differences, in total polyphenolic content, were observed between control and biostimulant treated fruits. Finally, after 120 days of cold storage, it was observed a decrease of total polyphenol content of 5.8% and 17.2%, respectively, for the control fruits and for LG biostimulant treated fruits. On the other hand, fruits derived from the PEP and MA treatments showed an obvious preservation of the nutritional quality, in terms of polyphenol content, not showing significant variations as compared to the fruits cold stored for 60 days (Table 5). Moreover, fruits derived from PEP treatment have shown, after 120 days of cold storage, a significant higher content of total polyphenols and a higher concentration of phloretin xylo-glucoside and phloridzin (350.53 and 34.102 mg/kg dw, respectively) than those determined in the other samples.

Furthermore, qualitative investigations carried out by HRMS Orbitrap showed that this is due to an increased level of certain compounds that are probably released during the cold storage from their polymeric forms or from dietary fiber to which they are linked. In particular, this release has been highlighted, in the case of PEP, for procyanidins, flavanols, and hydroxycinnamic acids, which underwent an increase, between 60 and 120 days of cold storage, accounting for 5–113%. Moreover, MA treatment caused, at the same time, an enhancement of flavonols between 0.6–28%. With respect to the flesh (Table 5), the reddening treatment did not cause statistically significant increases (*p* < 0.05) in extractable polyphenols, with the exception of the fruits deriving from the LG biostimulant treatment, which have a significant higher polyphenol content than the other analyzed samples (0.59 at harvest time and 0.92 after reddening), and also the highest concentration of hyperoside (19.68 vs. 4.90, 4.48 and 3.97 mg/Kg dw). In agreement with peel results, cold storage determined, after 60 days, an increase of polyphenol level for all of the samples, in particular there was an enhancement of 55%, 44.4%, 36.9%, and 37.8%, respectively, for control and PEP, LG, and MA biostimulant treated fruits. This finding was a consequence of a greater abundance of some individual polyphenols, such as phloretin xylo-glucoside, hyperoside, phloridzin, and kaempferol-3-O-glucoside, the concentration of which was two or three times higher than that found in the flesh of fruits after “melaio”. Furthermore, fruits derived from LG treatments, after 60 days of cold storage, showed a significant higher content of total polyphenols and a higher concentration of procyanidins (164.77 mg/kg dw) than those that were determined in the other samples. The long-term cold storage (120 days) satisfactorily maintained phenolic content and even fruits derived from MA and PEP application, in accordance with data obtained for peel, showed an increase of 7.8 and 5.8%, respectively, when compared to the fruits that were cold stored for 60 days.

These findings were supported by HRMS-orbitrap data, which showed, between 60 and 120 days of cold storage, an increase of flavanols and procyanidins, for fruits deriving from PEP treatment, of 11–56%, while MA application determined, mainly, an increase of flavonols, especially hyperoside and isorhamnetin glucoside, which increased from 10 to 19.7 and from 0.371 to 1.58 mg/Kg dw, respectively.

### 2.4. Effect of Biostimulants on Antioxidant Capacity

The radical scavenging of the apple extract (peel and flesh, separately) was determined using two well-known spectrophotometric assays by determining DPPH free radical-scavenging activity and ferric reducing antioxidant capacity; the results are reported in Table 5 and expressed as mmol trolox/kg dw. A calibration curve of inhibition, built with trolox^®^, was employed as a positive control for both essays. In the case of DPPH (Table 5), the antioxidant capacity ranged from 7.21 to 36.53 mmol trolox/kg dw for the different apple samples (peel, panel A and flesh, panel B). Overall, the antioxidant capacity was higher in the peel than in the flesh with a significant increase (*p* = 0.05), for peels after ripening, when PEP and LG treatment were applied.

During the cold storage, a decrease in the antioxidant capacity was found for all of the samples. However, it should be noted that the MA treatment led to a significant (*p* = 0.05) increase of peel antioxidant capacity after long-term cold storage (120 days). With respect to the flesh, the antioxidant capacity, did not show neither remarkable changes related to the ripening process or positive effects associated with the application of biostimulants. On the contrary, during cold storage, in accordance with total polyphenolic content measured by Folin assay, all of the apple samples, showed a significant increase in antioxidant capacity (*p* = 0.05), without any significant effect of the treatments compared to the untreated sample.

In terms of antioxidant capacity evaluated by FRAP assay, the values ranged from 11.86 to 89.91 mmol trolox/kg dw for the different apple samples and it was consistently higher in the peel than in the flesh. With regard to the peel (Table 5), the results highlighted a significant (*p* = 0.05) increase in the antioxidant capacity after ripening with the notable exception of the sample treated with microalgae based biostimulant, for which, was observed a decrease of 16.5% compared to the treated unripe fruit. In accordance with the DPPH data, after ripening, PEP and LG treatments led to a significant ameliorating effect on the antioxidant capacity when compared to the untreated fruit (61.98 vs. 54.40 mmol trolox/kg dw). Unlike the DPPH data, a significant (*p* = 0.05) increase in the peel antioxidant capacity was observed during the cold storage. However, after 60 days, no significant effect linked to the application of biostimulants was observed. The long-term cold storage (120 days) caused a significant decrease of antioxidant capacity for the control and for fruits produced by the plants that were treated with LG biostimulant. On the other hand, PEP treatments showed no significant changes, while MA based biostimulant contributed to a significantly higher antioxidant capacity with respect to untreated peel fruit (89.91 vs. 78.61 mmol trolox/kg dw), in accordance with DPPH data. With regard to the flesh (Table 5), after ripening, according to DPPH data, neither clear increases in antioxidant capacity nor improvement effects associated with the application of biostimulants have been observed. A definite positive effect of storage was detectable for flesh antioxidant capacity that was measured with FRAP assay. In particular, there is a considerable increase during storage (60 days), with a significant effect of LG biostimulant, which showed the highest antioxidant capacity (34.41 mmol trolox/Kg dw; +14.3% as compared to fruits produced in non-treated control plants).

These data were in accordance with the total polyphenolic content that was determined by Folin assay and with DPPH data. The FRAP values remained roughly constant during storage without remarkable variation at four months. Finally, we point out that LG and MA biostimulants significantly enhanced the FRAP values after 120 days of cold storage (+7.5 and +5.4%, respectively, as compared to untreated fruit). Therefore, on the basis of our data, treatments with PEP and LG were able to significantly (*p* = 0.05) increase DPPH and FRAP antioxidant capacity in the skin tissue of annurca apples, during reddening in “melaio”, whereas microalgae based biostimulant enhanced the antioxidant potential, especially at the skin level, when long-term storage was considered.

These results are consistent with those of Folin, with there being a good linear correlation (data not shown) found between the total polyphenols content and the antioxidant capacity measured with both DPPH and FRAP.

Principal component analysis (PCA) was conducted to characterize each biostimulant treatment with respect to antioxidant activity and total polyphenolic content. Three principal components (PCs) with eigenvalues greater than 1 and explaining 99.6% of the total variance of the data were obtained with PC1, PC2, and PC3, accounting for 50.8%, 29.8%, and 19.0%, respectively (data not shown). Figure 3 shows the bidimensional representation defined by the first and second PCs for all of the variables and biostimulant treatments investigated.

The plot describes the correlation between the variables highlighting that LG treatment was well related with flesh total polyphenols after redding, FRAP, DPPH, and Folin results after 60 days of cold storage and with the FRAP results obtained from flesh cold stored for 120 days. On the other hand, PEP treatment was found to be related to FRAP, DPPH, and Folin results obtained from apple flesh at harvest and with total phenols measured on cold stored peels for 120 days. Finally, MA treatment was well related to FRAP and DPPH peel results at harvest and interestingly after 120 days of cold storage (Figure 3).

## 3. Discussion

Apple represents an important source of positive substances for human health and the beneficial activity of these compounds is due to the antioxidant capacity of these compounds that belong to the class of polyphenols. The main classes of polyphenols in apple are flavonoids, dihydrochalcones, and other polyphenolic compounds, such as chlorogenic acid. The annurca apple is a widespread apple cultivar of southern Italy, which generally undergoes a reddening treatment after the harvest. To encourage the development of eco-sustainable agriculture, fruit and vegetable operators are increasingly applying organic agricultural techniques. Frequently, the adoption of this production system determines a significant reduction in production yield, making it necessary to use natural substances that, in some way, promote productivity and, at the same time, determine an improvement in the nutritional quality of the fruits. In the present study, the effects of three different biostimulants (Micro-algae (MA), Protein hydrolysate (PEP), and Macro-algae mixed with zinc and potassium (LG)) on primary apple quality traits and potential nutraceutical compounds at harvest and during storage were evaluated. The application with biostimulants has positive effects on the peel and fruits coloring, as reported in the literature [23]. Soppelsa et al. 2018 [24] have reported that biostimulants that are based on seaweed, B-group vitamins, and protein hydrolysate had a limited effect on primary apple quality traits (total sugar, acidity, size, and firmness), whereas they were able to significantly increase the intensity and extension of the red coloration of “Jonathan” apples at harvest. This could be probably attributed to a modulation of pathways for the biosynthesis of phytohormones in plants (mainly cytokines and abscisic acid), which are involved in the control of anthocyanins biosynthesis in apple skin before harvesting [25].

The total soluble solids (TSS) content is an important character that indicates the evolution of post-harvest maturation and in this study (Table 2), in accordance with Robert and von Loeska [26], an increase in TSS from harvest to fridgeconservation was observed. Contrary to TSS, as reported in the literature, after “melaio”, the Annurca fruits show a significant reduction in the acidity and firmness of the pulp [27]. Our results confirm Crouch’s data (2003) [28], which reported a decline in titratable acidity in the apple fruit after harvest. Several studies show that one of the main factors that contribute to the firmness pulp are the pectic substances that contribute to the adhesion between the cells, acting as a stabilizing gel [29,30]. The decrease in the firmness of Annurca apples after reddening could be due to an increase of large polymers in K oxalate-dimethylsulphoxide soluble pectins mainly composed of galacturonic acid and also the decrease of low-molecular-weight fragments composed of glucose and rhamnose [15].

The classes of polyphenolic compounds that were found in the apple extracts were common to all investigated apple type and in agreement with those reported in previous studies on this cultivar [7,31,32]. Polyphenolic compounds were extracted using the method described in the literature [33], where it is reported that the most satisfactory extraction efficiency was achieved extracting polyphenolic compounds from lyophilized apple peel and pulp, with methanol/water (90:10, *v/v*) mixture employing an ultrasound bath.

The main compounds are chlorogenic acid, hyperoside, catechin, epicatechin, procyanidins, phoridzine, and phloretin xylo-glucoside. The total phenolic amounts obtained from the UHPLC-HRMS analysis was slightly lower and not well correlated from that estimated by the Folin–Ciocalteau method. This result is abundantly explained in the literature [34] and, in this case, attributed to the fact that Folin assay allows for an overall determination of bioactive compounds, including high-molecular-weight phenolic compounds that have not been investigated in the mass spectrometry analysis, such as tannins or polymeric procyanidins.

The application of seaweed extracts, or their components, to food crops is widely used and provides benefits to the health consumers. Seaweeds are rich sources of macro- and micro-elemental nutrients, amino acids, vitamins, and compounds, which may have effects on enhancing the nutritional value of the treated plants. Studies on the effect of *A. nodosum* extract treatment to spinach (Spinacia oleracea L.) showed that the treatment improved postharvest storage quality and, at the same time, enhanced flavonoid content in spinach leaves [35]. Lola-Luz et al. 2014 [36] indicated that there was an increase in flavonoid and phenolic compounds following seaweed extract application in potatoes and onion. Soppelsa et al. 2018 [24] reported that total phenolic content (TPC) evaluated at the skin level, in Jonathan apples, was significantly affected by macroseaweed treatment, highlighting that this biostimulant significantly enhanced total polyphenols in apple skin, while a not significant effect was observed when a mix of amino acids was used as biostimulant. In the same study, higher phenolic concentration and antioxidant capacity were also detected in apple skin after B-group vitamins application, attributing this effect to the ability of thiamine to elicite various genes that belong to the phenylpropanoid pathway with consequent higher increase of secondary metabolites and antioxidant capacity. The health promoting qualities of LG biostimulant can also be attributed to the zinc to which Soppelsa et al. 2018 [24] attributed an improving effect on the polyphenols total content and antioxidant capacity in apple. Tamas et al. 2019 [37] investigated the effect of algae based biostimulants on nutritional quality of apples cv. Gala Must and found that the applied products significantly increased the amount of flavonoid, phenolic compounds and antioxidant capacity. In the literature, several studies described the use of protein hydrolysates, in apples and grapes, or only phenylalanine in grapes and highlighted the positive effects on the total polyphenol and anthocyanin contents [24,38,39]. Our results were different from those that were reported by Soppelsa et al. 2018 [24], which showed that the total polyphenolic content at the pulp level was not significantly affected by the treatments with several biostimulants, including, among others, macro and micro seaweed extracts.

In the literature, it was also reported that the health promoting effect of microalgae treatments on peel polyphenolic content of Jonathan apples [24], in our case microalgae treatments have not shown, for both the peel and for the flesh, significant effects on the content of total polyphenols but showed a positive effect during the cold storage, contributing to a greater release of single bioactive molecules from fiber-bounded and polymeric polyphenols. After 60 days of cold storage, an increase in peel total polyphenol content was observed for all the types of sample investigated, according to literature data which attribute the increase to the ethylene action which stimulate the biosynthetic pathway of phenol compounds [7].

One of the major objectives in agricultural production is the introduction of value-added qualities, especially nutritional characteristics. On this account, biostimulants have been described as a powerful tool for enhancing the nutritional properties of food crops [24]. In more detail, biostimulants can improve bioactive aspects of vegetable foods, by up-regulating a number of genes implied in the secondary metabolism, responsible for the synthesis of compounds, such as phenols and terpenes, which lead to the enhanced antioxidant capacity and the increased tolerance to biotic and abiotic stresses [40]. Few studies have shown that the exogenous application of biostimulants increased endogenous antioxidant capacity in apples, such as increased amounts of phenolics and anthocyanin content in apples that were treated with biostimulants. Our results were in accordance with the data illustrated by Soppelsa et al. 2018 [24] and Malaguti et al. 2002 [41], who reported, in organic apple of the cultivar Jonathan and Mondial Gala, similar health promoting responses regarding the significant improvement of antioxidant capacity following the use of macro and micro seaweed extracts both in the peel and in the pulp and the considerable increase of pulp antioxidant potential as a result of amino acids applications. High antioxidant activities that are related to the biostimulants treatment are fundamental for improving the nutritional value of fruit as well as extending their shelf life, thus increasing the overall quality and marketable value of fresh products [35].

## 4. Materials and Methods

### 4.1. Reagents and Materials

Polyphenolic standards, including rutin, catechin, epicatechin, chlorogenic acid, caffeic acid, procyanidin b1, procyanidin b2, phloridzin, kaempferol 3-O-glucoside, apigenin glucoside, and phloretin, were purchased from Sigma–Aldrich Chemical Co. (St. Louis, MO, USA). For the antioxidant tests, gallic acid, 6-hydroxy-2,5,7,8-tetramethylchromane-2-carboxylic acid (Trolox), 1,1-diphenyl-2-picrylhydrazyl (DPPH), 2,3,5-triphenyltetrazolium chloride (TPTZ), anhydrous ferric chloride, hydrochloric acid, and sodium acetate were purchased from Sigma–Aldrich (Milan, Italy). Methanol (MeOH) and water (LC-MS grade) were acquired from Carlo Erba reagents (Milan, Italy), whereas formic acid (98–100%) was purchased from Fluka (Milan, Italy).

### 4.2. Experimental Site and Plant Material

The experiment was conducted in 2019 in Vitulazio (CE) (41°9′54″36 N, 14°13′4″08 E, 57 m a.s.l.) at an apple orchard of Giaccio Frutta Cooperative Society, on cv Annurca plants, aged eight years, grafted on M9 rootstock. The plants were trained to spindle training systems and spaced 4.5 m between the rows and 1.5 m on the row with a planting density of 1481 trees ha^−1^, the cv Sergente was used as a pollinator. The experiment was carried out on medium-textured soil with an adequate content of macro and micro elements. Irrigation was provided using a drip system equipped with two self-compensating drippers for plant, delivering 8 L/h. The orchard received standard horticultural cares and the treatments against the main parasites have been established in accordance with the regulation governing integrated production.

### 4.3. Design and Biostimulant Treatments

The experiment set up was organized as a completely randomized block design with four replications per treatment and 10 trees per replicate, per a total of 40 pants per treatment. To avoid any contamination between treatments, replicates on the same row were separated by an interval of 10 untreated trees. The trees were selected according to uniformity of fruit load and vegetative activity. For treatments were used a backpack atomizer with internal combustion engine, model Geotech Pro MDP 500, the quantity of water used for each treatment was 50 L. The foliar applications were carried out starting from the fruit set with a time interval of about 10 days between one application and another. In particular, the treatments started on 4/07/2019, when the fruit diameter was 43 mm, and they were repeated on the following dates: 18/07, 6/08, 19/08, 29/08, 9/09, and 18/09. The execution of the manual thinning of the fruits was prior to the treatments.

Four treatments were compared:

Control (Control), plants not treated with biostimulant but only with water.Protein hydrolysate (PEP), the product used was Peptone 85/16 from A.I.CHEM company (Milan, Italy), authorized in organic farming, a biostimulant with a high concentration of amino acids. The product was applied by foliar application with 150 g/50 L of water.Meaweed mix(Ascophyllum nodosum and Laminaria digitate), a zinc-based formulation (ZINC 10 LG S) and a liquid potassium-based fertilizer (Red Skin LG). These products were mixed together and used by foliar application. The first two treatments were carried out with LG 201 + ZINC 10 LG S, while the following with Red Skin LG + seaweed mix with a dosage of 100 mL/50 L for LG 201 and LG74 while 150 mL/50 L for LG 347. The overall treatment was called LG.MicroAlgae (MA), the product used was AgriAlgae Biologico Original from AgriAlgae^®^ (Madrid, Spain), a high quality biological biostimulant that was obtained from micro-algae. The product was applied by foliar application with 200 mL/50 L of water.

### 4.4. Physico-chemical Analysis of Fruits

Each sample consisted of 40 fruits per treatment taken at the harvest on 13/09, after “melaio” and after 60 and 120 days of refrigeration at +2 °C, the parameters analyzed were: weight, firmness pulp, epicarp coloring (using color coordinates L, a, b CIELAB), iodine-iodide test for the determination of starch index, total soluble solids (TSS) content, pH, and titratable acidity (TA). The weight was determined with an electronic scale, while the pulp firmness with an EFFEGI manual penetrometer with an 8 mm tip on two sides opposite the fruit. The TSS content was determined with a HI 96,814 digital refractometer of Hanna instruments. TSS or Brix represents the percentage by mass of total soluble solids of a pure aqueous sucrose solution [42]. The pH was determined with a pH meter by the Hanna Instruments laboratory and total acidity with an acid-base titration. The solution was titrated with 0.1N sodium hydroxide standard solution.

Fruit epicarp coloring was determined with a colorimeter (Minolta, model CR-400, Tokyo, Japan) that was capable of quantifying colors according to international standards and expressed in defined color spaces. The instrument was calibrated with “white” managed by the light source on a white tile, before each measurement. The L * a * b * (CIELAB) color space is the most common for measuring the color of an object or materials of different origins and it is widely used in all sectors. In this color space, L * indicates brightness, while a * and b * the chromaticity coordinates: +a * is the direction of red, −a * is the direction of green, +b * is the direction of yellow, and −b * is the direction of blue [43]. The measuring was repeated four times in different points of the fruit, on the face not exposed to the sun, on the face exposed to the sun, and on the two intermediates, in order to determine the chromatic parameters of the fruit. Subsequently, the colorimetric index (IC) was calculated using the formula: IC = (1000 × a)/(b × L).

### 4.5. Polyphenols Extraction and Analysis by UHPLC-Q-Orbitrap HRMS

Polyphenols were extracted according to the method of Petkovska et al. 2016 [33], with few modifications. Briefly, 0.5 g of pulp/peel was extracted with two different portions (2.5 mL) of methanol: water (80:20 *v/v*), the mixture was vortexed intensively for 1 min. and sonicated in the dark, at room temperature, for 30 min. After centrifugation (3000 rpm/min.) for 10 min. the supernatants from both extractions were combined and made up to a final volume of 5 mL. The extracts were filtered through 0.22 µm nylon filters (Phenomenex, Castel Maggiore, Italy), prior to injection into the UHPLC-Orbitrap MS. The same extracts were used for antioxidant capacity and total polyphenolic content determinations. Chromatographic analysis was performed through an UHPLC system (UHPLC, Thermo Fisher Scientific, Waltham, MA, USA), equipped with a Dionex Ultimate 3000 Quaternary pump and a thermostated (25 °C) Kinetex 1.7 µm biphenyl (10 × 2.1 mm) column (Phenomenex, Torrance, CA, USA), with the following analytical conditions: solvent A, water/formic acid (99.9:0.1); solvent B, methanol/formic acid (99.9:0.1). Flow rate, 0.2 mL/min; injection volume, 2 µL. The autosampler and column temperatures were set at 10 °C and 25 °C, respectively. A gradient elution program was applied, as follows: 0 min, 5% of phase B; 1.3 min, 30% of phase B; 9.3 min, 100% of phase B; 11.3 min, 100% of phase B; 13.3 min, 5% of phase B; 20 min, 5% of phase B. As some standards were not available, quantitation for some polyphenols was calculated employing calibration curves of structurally related substances that belong to the same chemical group and with a similar response.

The mass spectrometry analysis was facilitated by a Q Exactive Orbitrap LC-MS/MS (Thermo Fisher Scientific, Waltham, MA, USA) that was equipped with an electrospray (ESI) source operating in negative ion mode (Thermo Scientific, Bremen, Germany). The acquisitions were conducted by setting a Full MS/AIF mode that uses a full MS scan (without HCD fragmentation), followed by an all ion fragmentation (AIF) scan (with a fragmentation energy applied). Full MS experiments were carried out with settings: microscans, 1; AGC target, 1e6; maximum injection time, 200 ms; mass resolution, 35,000 FWHM at *m/z* 200, whereas the AIF scan conditions were: microscans, 1; AGC target, 1e5; maximum injection time, 200 ms; mass resolution, 17,500 FWHM at *m/z* 200; HCD energy, at 10, 20 and 45. In both cases, the instrument was set to spray voltage, 3.5 kV; capillary temperature, 275 °C; sheath gas, 45 (arbitrary units); auxiliary gas, 10 (arbitrary units); *m/z* range, 80–1200; data acquisition, profile mode. The accuracy of MS analysis was ensured by calibrating the detector using the commercial calibration solutions that were provided by the manufacturer. Mass tolerance was kept at 5 ppm in both fullscan MS and AIF modes. Xcalibur software v. 3.1.66.10 (Xcalibur, Thermo Fisher Scientific, v. 3.0.63) was used to perform data analysis and processing.

### 4.6. Determination of Total Phenolics

The fruit content of total phenolics was determined according to a Folin–Ciocalteu procedure [44], with slight modifications. Briefly, 125 µL of diluted extract or blank (125 µL methanol/water, 80:20 *v:v*) was mixed with 500 µL of deionized water and 125 µL of the Folin–Ciocalteu reagent for 6 min. at room temperature.

Subsequently, 1.25 mL of 7.5% of sodium carbonate solution and 1 mL of deionized water were added in the mixture. The absorbance at 760 nm after 90 min. of incubation in the dark was measured. Concentrations of total phenolic were expressed in terms of mg of gallic acid equivalents (GAE) per gram dry weight (DW), based on a standard linear curve (*R*^2^ > 0.995) that was computed over a dynamic range 0.05–2.5 g/L gallic acid. Each extract was analyzed in triplicate.

### 4.7. Determination of DPPH. Scavenging Activity

The 1,1-diphenyl-2-picrylhydrazyl (DPPH) free radical scavenging activity of apple extracts was determined using the procedure that was described by Brand-Williams et al. [45], with minor modifications. Briefly, methanolic DPPH^.^ (4 mg in 10 mL) was diluted with methanol to an absorbance value of 0.90 (±0.02) at 517 nm to obtain a DPPH radical working solution. The scavenging activity of the apple extracts was determined by adding l mL of DPPH radical working solution and 200 µL of suitably diluted apple extract. The decrease in absorbance of the resulting solution was monitored at 517 nm after 10 min. of incubation at room temperature in the dark. The results were corrected for dilution and expressed in TEAC (mmol Trolox equivalents per kg dry weight of sample). All of the determinations were performed in triplicate.

### 4.8. Determination of Ferric Reducing Antioxidant Capacity

Ferric-reducing antioxidant capacity (FRAP) assay was conducted based on the method of Benzie and Strain [46], with minor modifications. Briefly, FRAP reagent was made up of 1.25 mL of 10 mmol 2,4,6-tripyridyl-striazine (TPTZ) in HCL (40 mL), 1.25 mL of FeCl3 (20 mmol) in water, and 12.5 mL of acetate buffer (0.3 M, pH 3.6). The apple extracts (150 μL) were allowed to react with 2.850 mL of FRAP reagent. The absorbance was monitored after 4 min. at 593 nm. The results were expressed as TEAC (mmol Trolox equivalents per kg dry weight of sample). All of the determinations were performed in triplicate.

### 4.9. Statistical Analysis

Analysis of variance (ANOVA) on the complete randomized block design on the data and mean separation by Duncan’s multiple range test (*p* < 0.05) were performed while using the XLSTAT, version 2013, statistical software package (New York, NY, USA).

## 5. Conclusions

This study represents the first detailed research into the use of different type of biostimulants on annurca apple quality at harvest and during storage. In the literature, there have not yet been reported data on the effects of biostimulant applications on annurca apple quality; therefore, the results obtained in this study appeared essential when considering that the biostimulant action can vary, depending on species/cultivar [47].

These results suggest that selected biostimulants are involved in regulating the secondary metabolism of treated plants, leading to an improvement of annurca fruit quality and nutritional value. These beneficial qualities are also combined with their key role, in dealing sustainability challenges, reducing dependency on chemical fertilizers that are increasingly expensive due to resource depletion, and growing global demand and dangerous to human health and the ecosystem.

The qualitative/quantitative improving effect on polyphenolic profile of fruits as well as the enhancement of their antioxidant capacity following biostimulant application are relevant factors to further improve its well-known nutraceutical potential, conservation, and commercialization.

## Figures and Tables

**Figure 1 plants-09-00775-f001:**
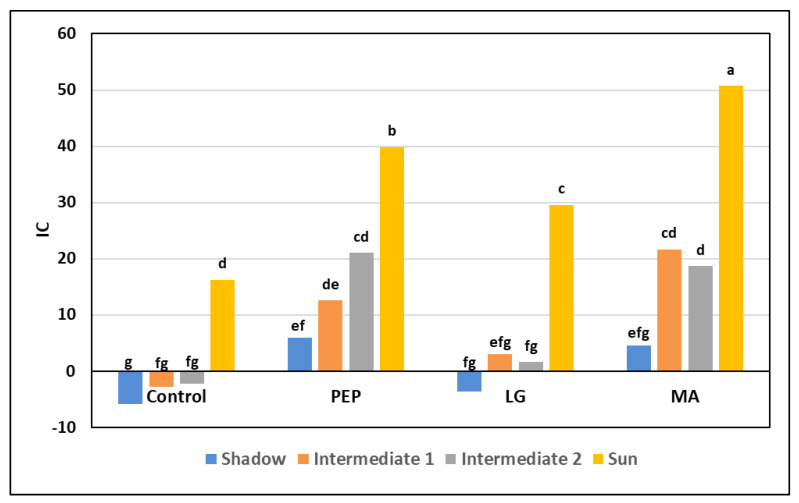
Effect of biostimulants on colorimetric index (IC) of Annurca fruits with different exposure at harvest. Same letter indicates not significant differences according to Duncan’s multiple range test (*p* < 0.05).

**Figure 2 plants-09-00775-f002:**
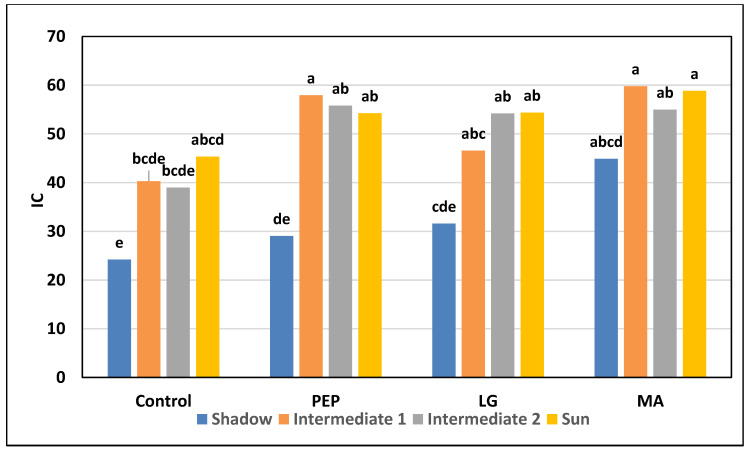
Effect of biostimulants on colorimetric index (IC) of Annurca fruits with different exposure after redding. Same letter indicates not significant differences according to Duncan’s multiple range test (*p* < 0.05).

**Figure 3 plants-09-00775-f003:**
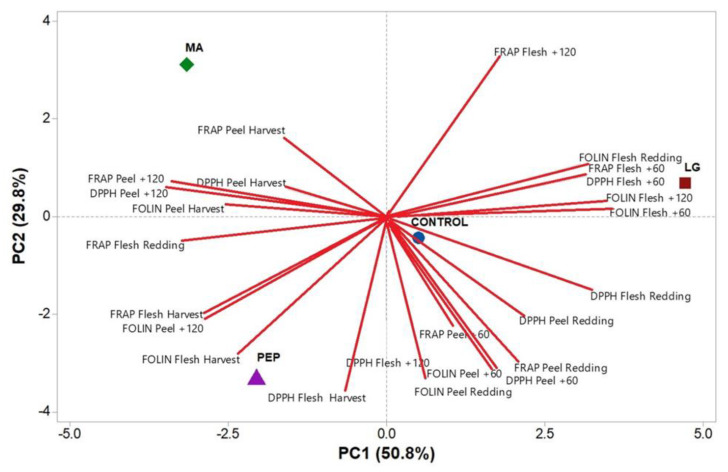
Principal component analysis (PCA) based on the biostimulant treatments with respect to antioxidant activity (ferric-reducing antioxidant capacity (FRAP) and 1,1-diphenyl-2-picrylhydrazyl (DPPH)) and total polyphenolic content (FOLIN) in peel and flesh of Annurca fruits.

**Table 1 plants-09-00775-t001:** Effect of biostimulants on Annurca fruit color (L* a* b*) with different exposure at harvest. ns, *, *** non-significant or significant at *p* ≤ 0.05 and 0.001, respectively. Different letters within each column indicate significant differences according to Duncan’s multiple-range test (*p* = 0.05). All of the data are expressed as mean ±SE, *n* = 20.

Source of Variance	L*	a*	b*
Biostimulant (B)			
Control	65.20 ± 0.87 a	-2.83 ± 1.73 c	32.33 ± 0.75 a
PEP	53.87 ± 1.31 c	12.91 ± 1.75 a	24.70 ± 1.02 c
LG	60.79 ± 1.09 b	2.07 ± 1.76 b	27.96 ± 0.97 b
MA	52.29 ± 1.16 c	14.94 ± 1.68 a	22.61 ± 0.89 d
Exposure of fruit (E)			
Shadow	66.06 ± 0.97 a	−6.81 ± 1.52 c	33.45 ± 0.82 a
Intermediate 1	59.45 ± 1.07 b	5.00 ± 1.65 b	27.71 ± 0.88 b
Intermediate 2	58.73 ± 1.10 b	5.21 ± 1.58 b	27.35 ± 0.89 b
Sun	47.91 ± 0.99 c	23.69 ± 1.09 a	19.10 ± 0.66 c
B × E			
Control × Shadow	71.63 ± 0.38 a	−15.55 ± 0.40 g	37.31 ± 0.56 a
Control × Intermediate 1	66.49 ± 1.15 abc	−7.26 ± 1.81 ef	33.82 ± 0.93 abc
Control × Intermediate 2	67.40 ± 0.87 abc	−6.00 ± 2.06 def	34.24 ± 1.06 ab
Control × Sun	55.29 ± 1.68 d	17.49 ± 3.24 bc	23.97 ± 1.29 f
PEP × Shadow	62.20 ± 2.64 c	−1.44 ± 3.82 de	32.11 ± 2.23 bcd
PEP × Intermediate 1	56.19 ± 2.12 d	10.92 ± 2.58 c	25.53 ± 1.59 ef
PEP × Intermediate 2	52.33 ± 2.18 de	15.01 ± 2.35 bc	23.16 ± 1.63 f
PEP × Sun	44.76 ± 2.01 fg	27.15 ± 1.41 a	18.01 ± 1.29 h
LG × Shadow	68.14 ± 1.12 ab	−10.86 ± 1.80 fg	34.27 ± 1.32 ab
LG × Intermediate 1	63.21 ± 1.91 bc	−0.94 ± 2.69 de	29.17 ± 1.91 de
LG × Intermediate 2	62.44 ± 1.54 c	−2.17 ± 2.62 de	29.56 ± 1.51 cde
LG × Sun	49.38 ± 1.55 ef	22.24 ± 1.69 ab	18.83 ± 1.12 gh
MA × Shadow	62.30 ± 1.95 c	0.59 ± 3.33 d	30.11 ± 1.63 bcd
MA × Intermediate 1	51.89 ± 1.65 de	17.29 ± 2.94 bc	22.30 ± 1.41 fg
MA × Intermediate 2	52.75 ± 1.85 de	13.99 ± 2.56 c	22.42 ± 1.50 fg
MA × Sun	42.22 ± 1.39 g	27.90 ± 0.88 a	15.61 ± 0.88 h
Significance			
Biostimulant (B)	***	***	***
Exposure of fruit (E)	***	***	***
B × E	ns	*	ns

**Table 2 plants-09-00775-t002:** Effects of biostimulants on total soluble solids (TSS) content, total acidity (TA), pH, and flesh firmness of Annurca fruits at harvest, after redding, and during the cold storage (+60 and +120 days) (*n* = 20 fruits). Mean values in the same row followed by different letters indicate significant differences (*p* > 0.05) using the Duncan’s multiple range test.

*Treatments*	*TSS (°Brix)*	*TA* (g∙L^−1^)	*pH*	*Firmness* (kg∙cm^2^)
***Harvest***				
Control	11.50 ± 0.12 de	9.60 ± 0.21 a	3.29 ± 0.08 f	-4.92 ± 0.17 ef
PEP	11.03 ± 0.24 ef	9.20 ± 0.36 a	3.41 ± 0.04 e	-4.82 ± 0.19 e
LG	11.46 ± 0.28 de	9.70 ± 0.15 a	3.32 ± 0.04 ef	-5.13 ± 0.16 ef
MA	11.27 ± 0.29 def	9.47 ± 0.03 a	3.42 ± 0.07 e	-5.46 ± 0.10 f	
***After redding***				
Control	11.73 ± 0.18 cd	5.87 ± 0.30 cd	3.61 ± 0.04 d	-3.13 ± 0.16 cd
PEP	10.77 ± 0.29 f	5.73 ± 0.23 cd	3.60 ± 0.02 d	-2.93 ± 0.19 abcd
LG	11.23 ± 0.17 def	5.67 ± 0.33 cd	3.60 ± 0.01 d	-3.00 ± 0.20 bcd
MA	11.33 ± 0.12 def	5.47 ± 0.12 cd	3.56 ± 0.06 d	-3.48 ± 0.08 d
***+60***				
Control	13.06 ± 0.03 b	5.57 ± 0.07 cd	4.01 ± 0.02 a	-2.69 ± 0.16 abc
PEP	11.70 ± 0.35 cd	5.90 ± 0.02 c	3.98 ± 0.06 a	-2.72 ± 0.08 abc
LG	11.57 ± 0.18 de	6.00 ± 0.15 c	3.68 ± 0.04 cd	-2.75 ± 0.11 abc
MA	11.63 ± 0.22de	5.93 ± 0.15 c	3.76 ± 0.03 bc	-2.77 ± 0.11 abc
***+120***				
Control	13.23 ± 0.03 b	5.20 ± 0.10 de	3.80 ± 0.03 b	-2.36 ± 0.09 a
PEP	13.27 ± 0.07 b	4.63 ± 0.03 e	3.85 ± 0.03 b	-2.38 ± 0.16 ab
LG	14.10 ± 0.06 a	5.53 ± 0.15 cd	3.80 ± 0.03 b	-3.05 ± 0.12 cd
MA	12.27 ± 0.03 c	5.57 ± 0.30 cd	3.78 ± 0.02 bc	-2.82 ± 0.06 abc

**Table 3 plants-09-00775-t003:** Effect of biostimulants on contents of phenolic compounds in flesh and peel of Annurca fruits at harvest, after redding, and during the cold storage (+60 and +120 days). Significance level of total polyphenols in Annurca apple peel in ANOVA test: ns: not significant; * 0.01 < *p* <0.05; ** 0.001 < *p* <0.01; ** *p* <0.001.

Flesh																			
*Polyphenols*	Control				PEP				LG				MA				*Statistical Significance*		
	*Harvest*	*After Redding*	*+ 60*	*+ 120*	*Harvest*	*After Redding*	*+ 60*	*+ 120*	*Harvest*	*After Redding*	*+ 60*	*+ 120*	*Harvest*	*After Redding*	*+ 60*	*+ 120*	*B*	*S*	*BxS*
procyanidin b1	49.278	96.303	54.365	66.008	106.003	81.585	41.519	55.790	49.242	115.751	67.442	68.509	75.572	112.264	63.812	65.538	***	***	***
catechin	83.705	123.266	48.054	55.820	109.862	81.150	41.411	64.673	134.461	118.707	60.773	75.869	95.416	120.871	63.614	66.180	*	***	ns
chlorogenic acid	754.459	861.785	410.505	368.963	786.097	812.944	380.849	380.000	443.596	829.350	373.970	387.205	875.085	825.864	403.150	372.827	ns	***	*
caffeic acid	0.199	0,000	0,000	0,000	0.032	0.037	0.000	0.000	0.065	0.059	0.000	0.000	0.066	0.025	0.000	0.000	**	***	***
procyanidin b2	88.537	164.735	63.923	71.726	139.559	121.385	54.136	60.071	68.042	151.315	72.864	69.863	122.572	156.358	69.624	67.494	***	***	***
epicatechin	202.944	361.561	168.348	177.825	215.223	205.014	146.822	208.469	181.532	255.942	183.297	225.046	242.087	328.794	208.767	208.860	***	***	***
coumaroyl quinic acid	41.566	70.268	8.775	4.797	39.135	29.043	7.330	5.248	112.844	57.393	4.843	7.044	59.515	57.332	10.865	5.577	***	***	***
rutin	0.621	0.546	2.093	2.064	0.632	0.261	2.653	1.973	0.602	0.758	3.282	2.165	0.623	0.316	1.930	2.548	***	***	***
phloretin xylo-glucoside	65.162	125.170	338.078	251.737	87.972	126.610	334.335	350.531	154.505	113.077	254.948	236.411	185.856	120.064	368.755	304.944	***	***	***
hyperoside	10.518	4.905	17.379	14.367	14.501	4.485	16.403	14.846	2.527	19.684	17.209	8.913	6.858	3.956	10.003	19.771	**	***	***
phloridzin	8.157	10.055	47.128	31.477	8.719	9.505	43.585	34.102	11.057	11.122	37.783	33.768	14.459	10.063	38.876	25.482	***	***	***
kaempferol-3-O-glucoside	6.800	5.056	9.893	11.870	7.544	4.113	10.747	7.760	6.361	9.510	14.469	8.027	5.496	4.894	7.332	5.752	***	***	***
apigenin-7-glucoside	0.030	0.141	0.014	0.017	0.072	0.026	0.010	0.003	0.060	0.093	0.085	0.007	0.027	0.037	0.000	0.007	***	***	***
phloretin	0.501	0.574	0.407	0.409	0.532	0.545	0.413	0.367	0.556	0.595	0.123	0.808	0.693	0.552	0.650	0.676	***	***	***
epicatechin trimer	12.612	30.599	18.161	19.855	19.175	15.874	15.566	14.316	12.245	24.981	21.404	15.607	18.670	21.216	20.045	17.508	***	***	***
epicatechin tetramer	1.244	2.843	2.547	2.574	1.584	1.349	1.928	1.600	0.816	2.656	3.174	0.904	1.626	2.405	2.015	1.803	***	***	***
isorhamnetin glucoside	1.800	0.738	0.956	1.118	2.390	0.651	1.342	1.078	0.609	2.575	1.356	1.510	1.111	0.570	0.371	1.583	***	***	***
isorhamnetin derivative	0.192	0.000	0.000	0.000	0.193	0.000	0.000	0.000	0.000	0.201	0.000	0.000	0.188	0.000	0.000	0.000	***	***	***
Total polyphenols	1328.325	1858.545	1190.626	1080.627	1539.224	1494.577	1099.049	1200.826	1179.119	1713.771	1117.022	1141.657	1705.921	1765.578	1269.809	1166.550	**	***	***
**Peel**																			
***Polyphenols***	**Control**				**PEP**				**LG**				**MA**				***Statistical significance***		
	***Harvest***	***After redding***	***+ 60***	***+ 120***	***Harvest***	***After redding***	***+ 60***	***+ 120***	***Harvest***	***After redding***	***+ 60***	***+ 120***	***Harvest***	***After redding***	***+ 60***	***+ 120***	***B***	***S***	***BxS***
procyanidin b1	97.612	91.130	35.076	49.089	95.032	80.569	29.032	53.629	105.173	99.749	45.803	57.682	82.325	80.256	54.787	51.249	***	***	***
catechin	76.036	84.352	32.090	38.669	82.673	80.894	24.124	50.983	90.760	87.630	34.790	50.100	79.718	86.790	45.043	45.049	***	***	***
chlorogenic acid	420.255	476.678	202.678	201.679	437.396	488.000	182.224	214.132	560.354	545.378	199.132	221.872	469.396	505.704	218.635	177.283	***	***	***
caffeic acid	0.343	0.037	0.000	0.000	0.347	0.100	0.000	0.000	0.520	0.176	0.000	0.000	0.546	0.131	0.000	0.000	***	***	***
procyanidin b2	297.238	262.311	77.054	80.859	271.622	202.230	80.447	84.079	281.432	234.798	88.385	85.421	249.868	173.268	90.726	86.189	***	***	***
epicatechin	379.426	342.643	170.128	176.647	380.727	297.161	160.645	198.362	414.974	353.901	186.615	194.501	368.345	298.211	207.596	203.853	***	***	***
coumaroyl quinic acid	21.762	28.211	10.175	8.328	24.643	25.959	8.303	12.955	68.831	34.088	9.684	11.233	31.710	30.314	12.417	12.323	***	***	***
rutin	28.567	66.062	86.842	85.268	35.806	63.538	105.804	75.929	51.278	34.010	83.870	56.809	53.163	30.966	76.824	77.281	***	***	***
phloretin xylo-glucoside	220.704	210.855	320.677	270.541	195.953	191.656	357.653	339.826	263.865	127.959	334.867	291.105	202.874	214.354	394.755	323.796	***	***	***
hyperoside	362.469	468.178	1085.833	1088.306	418.500	517.191	1019.497	864.834	580.080	333.740	914.992	557.853	573.752	464.432	1035.852	962.438	***	***	***
phloridzin	114.563	92.064	191.970	190.187	98.827	82.178	184.689	160.919	127.482	81.102	200.664	139.455	129.335	98.179	202.628	217.801	***	***	***
kaempferol-3-O-glucoside	99.208	87.309	134.987	131.142	103.338	109.917	140.614	86.982	105.754	87.487	133.885	113.089	138.708	96.464	104.463	134.142	*	***	***
apigenin-7-glucoside	0.809	1.119	0.723	0.758	1.164	1.159	0.845	0.618	0.870	1.064	0.781	0.475	1.897	1.092	0.549	0.700	***	***	***
phloretin	4.554	3.381	1.954	3.552	3.578	2.941	2.612	3.356	4.374	3.134	2.032	4.133	4.579	3.445	2.786	3.496	***	***	***
epicatechin trimer	66.634	60.538	28.033	26.746	62.625	50.755	30.231	29.749	76.096	62.431	34.258	24.337	67.521	38.869	39.446	31.739	***	***	***
epicatechin tetramer	8.461	8.216	3.548	3.615	11.143	6.735	4.038	4.061	8.320	8.208	4.372	2.629	8.431	5.983	4.468	2.523	***	***	***
isorhamnetin glucoside	111.424	72.586	213.946	279.190	147.095	125.524	245.350	132.893	74.443	84.782	244.837	201.639	246.682	91.940	193.306	221.762	***	***	***
isorhamnetin derivative	6.467	5.258	16.183	23.026	8.397	9.420	22.112	9.936	4.731	5.540	17.549	10.860	20.063	5.561	11.767	18.833	***	***	***
Total polyphenols	2316.189	2360.298	2611.947	2657.602	2378.866	2335.927	2598.220	2323.243	2819.337	2185.177	2536.516	2023.193	2728.913	2225.959	2696.048	2570.457	**	***	***

**Table 4 plants-09-00775-t004:** Retention time and exact mass spectra data of polyphenols investigated by UHPLC-HRMS Orbitrap.

Compounds	Molecular Formula	Theorethical Mass [M−H]^−^	Measured Mass [M−H]^−^	Err [ppm]	Tr (min)
procyanidin b1	C_30_H_26_O_12_	577.13515	577.13580	1.13	7.5
catechin	C_15_H_14_O_6_	289.07176	289.07224	1.66	7.65
chlorogenic acid	C_16_H_18_O_9_	353.0878	353.08798	0.51	8.13
caffeic acid	C_9_H_8_O_4_	179.03498	179.03455	−2.4	8.25
procyanidin b2	C_30_H_26_O_12_	577.13515	577.13550	0.61	8.31
epicatechin	C_15_H_14_O_6_	289.07176	289.07196	0.69	8.51
coumaroyl quinic acid	C_16_H_18_O_8_	337.09289	337.09338	1.45	9.39
rutin	C_27_H_30_O_16_	609.14611	609.14624	0.21	9.78
phloretin xylo-glucoside	C_26_H_32_O_14_	567.17193	567.17206	0.23	9.83
hyperoside	C_21_H_20_O_12_	463.0882	463.08500	−6.91	9.89
phloridzin	C_21_H_24_O_10_	435.12967	435.12961	−0.14	10.11
kaempferol-3-O-glucoside	C_21_H_20_O_11_	447.09328	447.09366	0.85	10.28
apigenin-7-glucoside	C_21_H_20_O_10_	431.09837	431.09869	0.74	10.67
phloretin	C_15_H_14_O_5_	273.07684	273.07755	2.6	11.21
epicatechin trimer	C_45_H_38_O_18_	865.19854	865.19928	0.86	8.74
epicatechin tetramer	C_60_H_50_O_24_	1153.26193	1153.26233	0.35	8.84
isorhamnetin glucoside	C_22_H_22_O_12_	477.10385	477.10440	1.15	10.47
isorhamnetin derivative	C_29_H_34_O_15_	621.14611	621.14667	0.9	10.74

**Table 5 plants-09-00775-t005:** Effects of biostimulants on total phenolic content (FOLIN) and on antioxidant activity (DPPH and ABTS) in peel and flesh of the Annurca fruits at harvest, after redding, and during the cold storage (+60 and +120 days). The same letter indicates not significant differences according to Duncan’s multiple range test (*p* < 0.05).

*Treatments*	*FOLIN*	*DPPH*	*FRAP*	*FOLIN*	*DPPH*	*FRAP*
	(mg/kg dw)	(mmol trolox/kg dw)	(mmol trolox/kg)	(mg/kg dw)	(mmol trolox/kg dw)	(mmol trolox/kg)
	***Harvest***					
	Peel			Flesh		
Control	1.995 g	27.721 fgh	45.400 h	0.693 g	7.797 gh	16.35 fg
PEP	2.461 cde	32.956 bc	55.084 g	0.952 ef	9.879 d	18.586 e
LG	2.154 fg	31.392 cd	53.610 g	0.591 h	7.839 gh	11.863 h
MA	2.514 bcde	33.602 b	61.400 f	0.707 g	7.210 h	15.797 g
	***After redding***					
	Peel			Flesh		
Control	2.132 g	29.319 ef	54.400 g	0.727 g	8.961 def	17.389 ef
PEP	2.523 bcde	34.792 ab	61.978 f	0.713 g	8.800 efg	17.323 ef
LG	2.315 ef	36.526 a	61.821 f	0.921 f	9.369 de	12.731 h
MA	2.018 g	28.061 fgh	44.505 h	0.744 g	8.230 fgh	17.586 ef
	***+ 60 days***					
	Peel			Flesh		
Control	2.734 ab	27.644 fg	84.154 b	1.128 b	13.053 a	30.112 cd
PEP	2.951 a	28.798 efg	82.119 bc	1.037 cd	11.574 c	29.691 d
LG	2.905 a	28.532 fgh	81.592 bc	1.255a	13.402 a	34.410 a
MA	2.600 bcde	26.579 h	79.908 cd	1.016 de	11.900 bc	30.094 cd
	***+ 120 days***					
	Peel			Flesh		
Control	2.588 bcd	27.090 gh	78.610 d	1.116 b	13.658 a	29.919 d
PEP	2.902 a	29.109 ef	84.189 b	1.103 bc	13.402 a	28.568 d
LG	2.399 de	29.160 ef	70.715 e	1.164 b	12.901 ab	32.164 b
MA	2.668 bc	30.6841 de	89.9087 a	1.099 bc	12.692 ab	31.498 bc

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
