# Peer review of "Effects of Biostimulants on Annurca Fruit Quality and Potential Nutraceutical Compounds at Harvest and during Storage"

_plants, 2020, doi:10.3390/plants9060775_

Round 1
Reviewer 1 Report
Interesting paper on the effect of commercial biostimulants in improving the color and quality of a local apple cultivar.
The introduction is appropriate.
In results:
Line 111, data in table 2 report a difference in increased sugars in comparison with the control only for the LG treatment and not also for PEP. Only data resulted significantly different should be evidenced.
Delete repetition of Table 2 legend (Lines 119-131).
In chapter 2.4 and overall: I suggest to replace and uniform the term antioxidant activity and antioxidant power with antioxidant capacity.
Correlation-PCA analyses on data from the Folin, DPPH, FRAP could help to better explain the variation of these factors in response to the different treatments.
Conclusions are supporting the important role of biostimulants in improving apple quality even in organic agriculture, to this some considerations have to be added in order the cost sustainability of these additional treatments and real consumer perception of the improved quality and absence of risks in the application of these new chemical products.
Author Response
Response to Reviewer 1 Comments
Point 1: Line 111, data in table 2 report a difference in increased sugars in comparison with the control only for the LG treatment and not also for PEP. Only data resulted significantly different should be evidenced.
Response 1: we agree with your remark, in the manuscript we reported only statistically significant data obtained for treatment with LG biostimulant.
Point 2: Delete repetition of Table 2 legend (Lines 119-131).
Response 2: the repetition has been deleted
Point 3: In chapter 2.4 and overall: I suggest to replace and uniform the term antioxidant activity and antioxidant power with antioxidant capacity.
Response 3: in the manuscript was used uniformly only the term antioxidant capacity
Point 4: Correlation-PCA analyses on data from the Folin, DPPH, FRAP could help to better explain the variation of these factors in response to the different treatments
Response 4: PCA analysis was added to better understand the interrelationship among and within the samples treated with different biostimulants (lines 276-287)
Point 5: Conclusions are supporting the important role of biostimulants in improving apple quality even in organic agriculture, to this some considerations have to be added in order the cost sustainability of these additional treatments and real consumer perception of the improved quality and absence of risks in the application of these new chemical products.
Response 5: In the manuscript were added considerations concerning the suggested aspects (lines 510-512):
“These beneficial qualities are also combined with their key role, in dealing sustainability challenges, reducing dependency on chemical fertilizers that are increasingly expensive due to resource depletion and growing global demand and dangerous to human health and the ecosystem”
Reviewer 2 Report
In Results section Authors should be focused only on Result. They have added sentences which should be in Discussion section for example sentence in line 191-192
Line 151-156 sentences should be moved to Material and methods section in paragraph concerning polyphenol compounds measurement
Sentence in lines 165-166 "Surprisingly, the amount of
165 total phenolics calculated from the data obtained by UHPLC-HRMS analysis was slightly lower and 166 not well correlated from that estimated by the Folin–Ciocalteau method" This is not surprise. Authors in discussion section should try explain why in UHPLC-HRMS analysis the concentration of polyphenolic compounds was lower. In literature are many articles which compares HPLS methods and method with the Folin–Ciocalteau reagent.
Additionally the methods of extraction should be also discusses how it effect polyphenols content.
Figures 3-5 should be changed in one table.
Author Response
Response to Reviewer 2 Comments
Point 1: In Results section Authors should be focused only on Result. They have added sentences which should be in Discussion section for example sentence in line 191-192
Response 1: We thank the referee for this remark. The sentence has been moved to the section related to the discussion of the results.
Point 2: Line 151-156 sentences should be moved to Material and methods section in paragraph concerning polyphenol compounds measurement
Response 2: The sentence:” As some standards were not available, quantitation for some polyphenols was calculated employing calibration curves of structurally related substances belonging to the same chemical group and with similar response” has been included in Material and methods section, in paragraph concerning polyphenol compounds measurement.
Point 3: Sentence in lines 165-166 "Surprisingly, the amount of
total phenolics calculated from the data obtained by UHPLC-HRMS analysis was slightly lower and not well correlated from that estimated by the Folin–Ciocalteau method" This is not surprise. Authors in discussion section should try explain why in UHPLC-HRMS analysis the concentration of polyphenolic compounds was lower. In literature are many articles which compares HPLS methods and method with the Folin–Ciocalteau reagent.
Response 3: the term “surprisingly” has been removed and in the discussion section the reason for this discrepancy has been explained. (line 317-322).
Point 4: Additionally the methods of extraction should be also discusses how it effect polyphenols content.
Response 4: this information has been added in the discussion section (line 312-315)
Point 5: Figures 3-5 should be changed in one table.
Response 5: the figures 3-5 have been replaced by the table 6., the caption has been changed. This correction has been made also in the manuscript.
Reviewer 3 Report
In general, the research reviewed here implies possible benefits of biostimulant application in horticultural production, especially in stressful growth conditions, such as the transplant stage, reduced fertilization, or incidence of other abiotic stress. Considering possible interactions among the contained physiologically-active compounds, the effects on plants may depend on dose, time of treatment, growth conditions, and plant species. Therefore, further research of biostimulant applications in horticultural production is suggested.
Research in this paper has shown:
This study represents the first detailed research into the use of different type of biostimulants on annurca apple quality at harvest and during storage. In literature have not yet been reported data on the effects of biostimulant applications on annurca apple quality, therefore results obtained in this study appeared essential considering that the biostimulant action can vary depending on species/cultivar. These results suggest that selected biostimulants are involved in regulating the secondary metabolism of treated plants, leading to an improvement of annurca fruit quality and nutritional value.
Research is interesting and useful in applying horticulture to science.
I am glad you chose a biostimulator. This is the future of fruits production.
Recommendation: Please, take a look at the new scientific paper:
Author Response
Thank you for your suggestions and considerations